# Trend and determinants of home delivery in Gambia, evidence from 2013 and 2020 Gambia Demographic and Health Survey: A multivariate decomposition analysis

Solomon Gedlu Nigatu*

Department of Epidemiology and Biostatistics, Institute of Public Health, College of Medicine and Health Sciences, University of Gondar, Gondar, Ethiopia

* sol.gondar@gmail.com

## Abstract

### Background

Home delivery is defined as is an even of pregnant women getting giving birth in a woman her home or other homes without an unskilled health professional assistance. It is continuing as public health problem since its responsible for death of women and newborn. In Gambia there is a high maternal mortality rate, which may be related to home delivery. Therefore, this study aimed to assess the trend of home delivery and identify predictors using Gambia Demographic and Health Survey (GDHS) 2013 and 2019–2020 data sets.

### Methods

A Cross-Section survey was conducted based on GDHS 2013 and 2019–2020 among reproductive age group women. A total of 8607 women participated in this study. A bivariate decomposition model was fitted, and variables that had a p-value > 0.25 were dropped. Finally, variables that got a p-value of < 0.05 with 95% confidence interval (CI) in the multivariate decomposition analysis were considered as statistical significance variables in the overall decomposition.

### Results

There has been a dramatic decrement in maternal home delivery in Gambia. It was 36.18% (95% CI:34.78, 37.58) in 2013 GDHS and 14.39% (95% CI:13.31,15.47) in 2019–2020 GDHS. This reduction is real because there was a change in the characteristics effect of the population and the coefficient effect some variables in the home delivery. Changes in characteristics effect of husband education, women education, rural residents, more than three antenatal cares follow up, and no problem reaching health facilities played a significant role in the reduction of home delivery. Being urban resident and women who had occupation were variables that had a positive effect on coefficient effect change.

**Data Availability Statement:** The Demographic and Health survey (DHS) program has conducted a survey of around 90 countries. It provides a data set freely available for every researcher. The only

requirement to access the data is online registration (www.dhsprogram.com).

**Funding:** The authors received no specific funding for this work.

**Competing interests:** The author has declared that no competing interests exist.

**Abbreviations:** ANC, Antenatal Care; EAs, Enumeration Area; GDHS, Gambia Demographic and health survey; LGA, Local Government Area; MMR, Maternal mortality Ratio; SDGs, Sustainable Development Goals; WHO, World Health Organization.

## Conclusion

In this study, the home delivery rate had steeply declined in the Gambia during the study period of the two surveys. Just above nine-tenths decrement in home delivery rate resulted because there was a change in the characteristics effect of the study participants. Enhancing more citizens to attend high school and above, narrowing the gap between rural and urban in terms of accessing health facilities, and improving the availability of infrastructure should be done.

## Background

Home delivery is defined as an even of pregnant women giving birth in her home or other homes without an unskilled health professional assistance [1, 2]. There is high inequality in home delivery across the globe, in high and middle-income nations only 1% home delivery had recorded whereas in low-income county 32% women experience home delivery [3]. In 2017, only 53% of women lived in Sub-Saharan Africa had institutional delivery [4]. A pooled prevalence from 11 East African countries revealed that home delivery was 23.68% [5].

Maternal death is one of the negative consequences of home birth [6]. Maternal mortality continues as a public health concern, 810 women dying every single day in 2017. These deaths occurred because of pregnancy and childbirth, which may have been avoided [7]. The lion's share (94%) of maternal deaths occurs in low and middle-income nations [8].

To save the lives of millions of women and newborns during childbirth, a professional birth attendant should be present. If nothing is done to modify the present number of childbirths supported by professional birth attendants, by 2030, nearly sixteen million newborns in Sub-Saharan Africa will be delivered without experienced birth attendants [9]. Another approach to preventing maternal mortality which is recommended by the World Health Organization (WHO) is every pregnant woman should have at least four antenatal care (ANC) visits before giving birth [10].

If a mother dies in a family, it has enormous negative consequences for the remaining family members. The first and most, infants whose moms died during or shortly after giving birth had a lower chance of survival than those whose mothers survived [11, 12]. Other consequences of a mother's death, particularly for children: It leads to malnutrition, school dropout, and an increased risk of mortality among children. Husbands will experience psychological related problems, and the remaining family member will separate [13–15].

Previous research has uncovered several factors that contribute to the decline of home delivery. To mention some of them: mother's age [16–18], place of residence [16, 19, 20], level of education of mother's [16, 18–20], wealth status [16, 19], marital status of mother's [16, 17], religion of mother's [19], number of antenatal care (ANC) [16–21], distance from health facility [8, 16].

Gambia has a high total fertility rate, a woman gave birth to an average of 4.4 children throughout her reproductive age. However, the Maternal Mortality Ratio (MMR) in Gambia was 289 women per 100,000 live births, which is a high proportion [22]. According to the Sustainable Development Goals (SDGs), MMR should not exceed 70/100,000 internationally, and the national MMR should not exceed double that of the global target. The MMR should be less than 140/100000 annually after 2030. Even though there are still years left, as the preceding data shows, Gambia's position to meet the SDG for women's health elucidates an enormous gap. It indicates that more work remains to be done to achieve these SDGs [19].

To the best knowledge of the author, there is scant information regarding the trend and determinants of home delivery in Gambia at a national level. Therefore, this research aims to assess the trend of home delivery in the Gambia and identify independent determinant factors.

## Materials and methods

### Study design and area

A cross-sectional study design was employed using the Gambia Demography and Health Surveys (GDHS) 2013 and 2019–20 data sets. Gambia is a country in West Africa, between latitudes 13 and 14˚N, and longitudes 13 and 17˚W. In 2020, the country's population was 2,416,668 people [1]. It is divided into eight Local Government Areas (LGA) for administrative purposes (Banjul, Kanifing, Brikama, Mansakonko, Kerewan, Kuntaur, Janjanbureh, and Basse). The Gambia health system is three levels: Primary, secondary, and tertiary. There are 4 referral hospitals, 8 main, and 16 small health centers across the nation [23].

### Data source, study population, and sampling technique

This study has used national representative secondary data collected in two different surveys. The source population for the study was all women in the reproductive age group (15–49 age) who gave birth 5 years preceding each survey. The study population was all women in the reproductive-age group who gave birth 5 years preceding each survey and lived in the designated enumeration areas (EAs). During both surveys, a stratified two-stage cluster sampling technique was utilized. Except for Banjul and Kanifing, which are entirely urban areas, the remaining LGA is divided into two categories: rural and urban. Exactly 281 EAs were chosen using the probability proportional technique. The next step was to utilize a systematic sampling procedure to select 25 households from each EA. The Selected women were questioned and responded to socio-demographic and obstetric factors. For the 2013 and 2019–2020 GDHS, the sample size was 4534 and 4073 women, respectively.

### Study variables

**Dependent variable.** The dependent variable was the place where most recent birth that happened in the last five years before the survey. It is a binary outcome, coded as 1 indicated the woman gave birth at her home or in another home, and 0 indicated that the woman gave birth in a health facility (government hospital, health center, health post, private hospital, and clinic).

**Independent variables.** Several independent variables were exhaustively included in this study: Women age (<20, 20–34,35–49), Residence (Urban, Rural), LAG(Banjul, Kanifing, Brikama, Mansakonko, Kerewan, Kuntaur, Janjanbureh, Basse), Religion (Muslim, Christianity), Women education (Unable to read and write, Primary, Secondary, Higher education), Husband education (Unable to read and write, Primary, Secondary, Higher education), Women occupation (Not working, Working), Wealth index (Poor, Middle, Rich), Had health insurance (No, Yes), Parity (<2, 2–5, >5), Birth order (1, 2–4, >4), Had ANC (<4, > = 4), and Health facility distance (Big problem, Not a big problem) (S1 File).

### Data management and analysis

After the data sets were downloaded from www.measuredhs.com then it was cleaned, recorded, and analyzed using STATA version 15.0 software. Different descriptive statistics were performed to describe the study participants using various variables including trends of home delivery with a 95% confidence interval (CI). The next step was to figure out which

predictor plays a significant role in the change in home delivery. The best technique to answer this research question is applying a non-linear multivariate logit decomposition analysis. For a long time, decomposition techniques have been applied for linear regression models, but now it is used for nonlinear models.

The first step in decomposition analysis is to decompose our dependent variable home delivery into a function of a linear combination of predictors and regression coefficients:

$$Y = F(X\beta)$$

where Y denotes the N × 1 Home delivery vector, X is an N × K matrix of independent variables, and β is a K ×1 vector of coefficients. F (·) is any once-differentiable function mapping a linear combination of X (Xβ) to Y. The mean difference in Y between groups A and B can be decomposed as

$$\overline{Y_A} - \overline{Y_B} = \overline{F(X_A\beta_A)} - \overline{F(X_B\beta_B)}$$
$$= \underbrace{\overline{F(X_A\beta_A) - F(X_B\beta_A)}}_{E} + \underbrace{\overline{F(X_B\beta_A) - F(X_B\beta_B)}}_{C}$$

The component mentioned above as E refers to the part of the differential attributable to differences in endowments or characteristics, usually called the explained component or characteristics effects. The C component refers to the part of the differential attributable to differences in coefficients (behavioral) or effects, usually called the unexplained component or coefficients effects.

The model structure of decomposition analysis is:

Logit (A) Logit (B) = $[B_{OA}-B_{OB}] +\sum B_{ijA} [X_{ijA}-X_{ijB}] + \sum X_{ijB}[B_{ijA}- B_{ijB}]$, where

- $B_{OA}$ is the **intercept** in the regression equation for GDHS 2019–20.

- $B_{OB}$ is the intercept in the regression equation for GDHS 2013.

- $B_{ijA}$ is the coefficient of the jth category of the ith determinant for GDHS 2019–20.

- $B_{ijB}$ is the coefficient of the jth category of the ith determinant for GDHS 2013.

- $X_{ijA}$ is the proportion of the jth category of the ith determinant for GDHS 2019–20.

- $X_{ijB}$ is the proportion of the jth category of the ith determinant for GDHS 2013.

A bivariate decomposition model was fitted, and variables that had a p-value > 0.25 were dropped. Finally, variables that get a p-value of < 0.05 with 95% CI in the multivariate decomposition analysis were considered as statistical significance variables in the overall decomposition (endowment and coefficient components) [24].

### Ethics approval and consent to participate

The study is based on secondary data, so it needed an email request from the DHS program's official www.measuredhs.com. The Gambia Bureau of Statistics and Ministry of Health (MoH) of Gambia collected the data by collaboration, and they received technical assistance from ICF through The DHS Program. The ethical clearance was granted by the Joint Ethics Committee: the Gambia Government/Medical Research Council (MRC) and institutional review boards (IRBs) at ICF international. Written consent from participates were obtained, and the information each study participants were recorded anonymously.

## Result

### Characteristics of the study population

In both surveys, more than two-thirds of the study participants were between the age of 20 to 34. During the study period, there was a significant shift in resident proportion, in which urban inhabitants increased from 47.9% to 63.7%. The majority of women (97.9%) identified as Muslim. Almost six out of ten women had two to five children in both surveys. The percentage of women who had no big problem reaching a health facility rose from 68.7% in 2013 to 73.3% in 2019–2020 GDHS (Table 1).

### Trends of home delivery during the survey period

Home delivery rate among women of reproductive age who gave birth in the last five years before each survey showed a significant decline over time. It was 36.18% [95% CI:34.78, 37.58] in Gambia in the first survey. After six years, another survey was conducting in 2019–2020, and home delivery rate had decreased drastically,. It was recorded as 14.39% [95% CI:13.31, 15.47] (Table 2). Several factors had played a key role in this decline. Basse LGA had the highest decline which was 46.45 rate of home delivery. In addition, there was a decrease in home delivery among women with poor and middle wealth index, they scored 28.18 and 34.17 rates of decline, respectively. However, families with health insurance showed the least home delivery decline rate and it was near one rate (Table 2).

### Decomposition of home delivery

After fitting multivariate decomposition logistic regression, it was clearly showing that 8.74% of home delivery rate decline has occurred for the reason of the change in characteristic effect (endowment) of the study pariciant. When we looked at the contribution of the coefficient effect for home delivery rate decline, it was attributed to 91.26% (Table 3). During the survey period, a little increase in the number of women who had secondary and higher education had a significant contribution to the reduction of the home delivery rate (1.29%). Even though there was a reduction in rural resident proportion in the sample, a 4.47% decrement in the home delivery rate was due to the characteristics effect changing of the study participants. Having a birth order two and above hampered the decrement in home delivery, it pulled back the decrement in the home delivery rate by 2.03% (Table 4).

An increase in the composition of women who had ANC follow of four times or more was a significant contributor to the decrement in the home delivery prevalence in the study period. There was an improvement in the number of women who accessed the health facilities in the study period (2013-2019/2020). The proportion of women who had no big problem accessed health facilities played a vital role in the decrement of home delivery prevalence, and its contribution was 1.77% (Table 4).

Home delivery rate decline occurred not only by the change in women's characteristics effect (endowment) but also by the coefficient effect of some variables. After keeping the contribution of endowment changes constant, a 91.26% home delivery decrement happened because of coefficient effect change. Almost half of the home delivery rate decline occurred for the fact that the change in women's health institution delivery coefficient effect among rural residents. Similarly, one-fourth of the decrement in home births was related to a change in women's health-care delivery coefficient effect among those who had a job. Although several factors contributed to the drop in home delivery rate over the study period, women who had more than four birth orders had an unfavorable impact on the trend of home delivery rate decrement.

**Table 1.  Percentage distribution of characteristics of respondents in 2013 and 2019–2020 Gambia Demographic and Health Surveys.**

| Characteristics of women | Category | Weighted frequency 2013 N = 4,534 | Percentage (%) 2013 | Weighted frequency 2019–20 N = 4,073 | Percentage (%) |
|---|---|---|---|---|---|
| Women age | <20 | 270 | 6 | 160 | 3.9 |
| | 20–34 | 3128 | 69 | 2696 | 66.2 |
| | 35–49 | 1136 | 25 | 1217 | 29.9 |
| Residence | Urban | 2174 | 47.9 | 2595 | 63.7 |
| | Rural | 2360 | 52.1 | 1478 | 36.3 |
| LAG | Banjul | 69 | 1.5 | 45 | 1.1 |
| | Kanifing | 751 | 16.6 | 672 | 16.5 |
| | Brikama | 1559 | 34.4 | 1620 | 39.8 |
| | Mansakonko | 240 | 5.3 | 183 | 4.5 |
| | Kerewan | 508 | 11.2 | 506 | 12.4 |
| | Kuntaur | 319 | 7 | 274 | 6.7 |
| | Janjanbureh | 397 | 8.8 | 303 | 7.5 |
| | Basse | 691 | 15.2 | 470 | 11.5 |
| Religion | Muslim | 4439 | 97.9 | 4002 | 98.5 |
| | Christianity | 95 | 2.1 | 71 | 1.8 |
| Women education | Unable to read and write | 2736 | 60.3 | 1983 | 48.7 |
| | Primary | 610 | 13.5 | 685 | 16.8 |
| | Secondary | 1044 | 23 | 1208 | 29.7 |
| | Higher education | 144 | 3.2 | 197 | 4.8 |
| Husband education | Unable to read and write | 2803 | 62 | 2240 | 55 |
| | Primary | 286 | 6.3 | 270 | 6.6 |
| | Secondary | 1135 | 25.1 | 1211 | 29.7 |
| | Higher education | 300 | 6.6 | 352 | 8.7 |
| Women occupation | Not working | 1880 | 41.5 | 1314 | 32.3 |
| | Working | 2654 | 58.5 | 2759 | 67.7 |
| Wealth index | Poor | 1884 | 41.6 | 1834 | 45 |
| | Middle | 931 | 20.5 | 852 | 20.9 |
| | Rich | 1719 | 37.9 | 1387 | 34.1 |
| Had health insurance | No | 4446 | 98.1 | 3975 | 97.6 |
| | Yes | 88 | 1.9 | 98 | 2.4 |
| Parity | <2 | 889 | 19.6 | 685 | 16.8 |
| | 2–5 | 2753 | 60.7 | 2546 | 62.5 |
| | >5 | 892 | 19.7 | 842 | 20.7 |
| Birth order | 1 | 814 | 18 | 632 | 15.5 |
| | 2–4 | 2103 | 46.4 | 1947 | 47.8 |
| | >4 | 1617 | 35.6 | 1494 | 36.7 |
| Had ANC | <4 | 994 | 21.9 | 806 | 19.8 |
| | > = 4 | 3,540 | 78.1 | 3266 | 80.2 |
| Health facility distance | Big problem | 1416 | 31.3 | 1086 | 26.7 |
| | Not a big problem | 3107 | 68.7 | 2987 | 73.3 |

**Table 2. Trends in home delivery rate among women's who gave birth in the last five years prior to the survey by selected characteristics, 2013 and 2019–2020 Gambia Demographic and Health Surveys.**

| Characteristics of women | Category | 2013 N = 4,533 | 2019–20 N = 12,037 | point difference in home delivery rate |
|---|---|---|---|---|
| Women age | <20 | 28.88 | 7.93 | -20.95 |
| | 20–34 | 36.85 | 14.21 | -22.64 |
| | 35–49 | 36.07 | 15.65 | -20.42 |
| Residence | Urban | 15.73 | 9.98 | -5.75 |
| | Rural | 55.02 | 22.13 | -32.89 |
| LAG | Banjul | 6.99 | 3.25 | -3.74 |
| | Kanifing | 11.03 | 8.05 | -2.98 |
| | Brikama | 22.17 | 10.75 | -11.42 |
| | Mansakonko | 44.40 | 29.02 | -15.38 |
| | Kerewan | 46.28 | 8.08 | -38.2 |
| | Kuntaur | 60.67 | 31.56 | -29.11 |
| | Janjanbureh | 48.59 | 22.62 | -25.97 |
| | Basse | 69.30 | 22.85 | -46.45 |
| Religion | Muslim | 36.48 | 14.60 | -21.88 |
| | Christianity | 22.25 | 2.38 | -19.87 |
| Women education | Unable to read and write | 44.25 | 19.44 | -24.81 |
| | Primary | 37.65 | 14.88 | -22.77 |
| | Secondary | 17.60 | 7.57 | -10.03 |
| | Higher education | 11.42 | 3.70 | -7.72 |
| Husband education | Unable to read and write | 45.73 | 19.29 | -26.44 |
| | Primary | 32.72 | 15.40 | -17.32 |
| | Secondary | 18.65 | 8.05 | -10.6 |
| | Higher education | 16.17 | 4.26 | -11.91 |
| Women occupation | Not working | 27.65 | 13.17 | -14.48 |
| | Working | 42.23 | 14.97 | -27.26 |
| Wealth index | Poor | 49.27 | 21.09 | -28.18 |
| | Middle | 48.74 | 14.57 | -34.17 |
| | Rich | 15.04 | 5.42 | -9.62 |
| Had health insurance | No | 36.76 | 14.67 | -22.09 |
| | Yes | 4.08 | 3.19 | -0.89 |
| Parity | <2 | 20.80 | 5.11 | -15.69 |
| | 2–5 | 39.04 | 15.22 | -23.82 |
| | >5 | 42.71 | 19.43 | -23.28 |
| Birth order | 1 | 19.63 | 4.75 | -14.88 |
| | 2–4 | 37.10 | 13.60 | -23.50 |
| | >4 | 43.33 | 19.50 | -23.83 |
| Had ANC | <4 | 45.61 | 20.36 | -25.25 |
| | > = 4 | 33.54 | 12.92 | -20.62 |
| Health facility distance | Big problem | 51.56 | 21.77 | -29.79 |
| | Not a big problem | 29.18 | 11.71 | -17.47 |
| Overall | | 36.18[34.78 37.58] | 14.39[13.31 15.47] | -21.79 |

**Table 3. Overall decomposition analysis of change in home delivery in Gambia, between 2013 and 2019–2020.**

| Home delivery | Coef. | [95% conf.interval] | Percent |
|---|---|---|---|
| E | -0.02054 | -0.02548–0.01559 | 8.74* |
| C | -0.21436 | -0.23205–0.19667 | 91.26* |
| R | -0.23490 | -0.25138–0.21842* | |

E: endowment; C: coefficient; R: residual,

*p-value< 0.001

**Table 4. Multivariate logistic regression decomposition analysis of home delivery in Gambia, 2013 and2019-2020.**

| Variable | | Difference in Characteristics (E) Coeff (95% CI) | Pct | Difference in Coefficients (C) Coeff (95% CI) | Pct |
|---|---|---|---|---|---|
| Women education | Unable to read and write | | | | |
| | Primary | 0.00016 (-0.00109 0.00141) | -0.07 | 0.00132 (-0.00644 0.00907) | -0.56 |
| | Secondary | -0.00207 (-0.00391–0.00023) * | 0.88 | 0.01094 (-0.00163 0.02352) | -4.66 |
| | Higher education | -0.00091(-0.00195 0.00013) | 0.39 | -0.00003(-0.00604 0.00610) | -0.01 |
| Husband education | Unable to read and write | | | | |
| | Primary | 0.00010(-0.00016 0.00035) | -0.04 | 0.00330 (-0.00150 0.00811) | -1.41 |
| | Secondary | -0.00209(-0.00292–0.00127) ** | 0.89 | -0.00257(-0.01579 0.01065) | 1.09 |
| | Higher education | -0.00093 (-0.00146–0.00040) ** | 0.40 | -0.00221(-0.00947 0.00504) | 0.94 |
| Religion | Muslim | | | | |
| | Christianity | 0.00017 (-0.00059 0.00093) | -0.07 | -0.00049 (-0.00443 0.00346) | 0.21 |
| Residence | Urban | | | | |
| | Rural | -0.01049(-0.01330–0.00769) ** | 4.47 | -0.11835 (-0.15411–0.08259) ** | 50.38 |
| Had health insurance | No | | | | |
| | Yes | -0.00006(-0.00025 0.00013) | 0.03 | 0.00353(-0.00122 0.00827) | -1.50 |
| Women age | <20 | | | | |
| | 20–34 | -0.00044 (-0.00253 0.00164) | 0.19 | 0.02609(-0.05745 0.10963) | -11.11 |
| | 35–49 | -0.00041 (-0.00382 0.00300) | 0.18 | 0.02252(-0.01176 0.05680) | -9.59 |
| Women occupation | Not working | | | | |
| | Working | -0.00133(-0.00439 0.00172) | 0.57 | -0.05898(-0.08620–0.03176) ** | 25.11 |
| Birth order | 1 | | | | |
| | 2–4 | 0.00075(0.00050 0.00101) ** | -0.32 | 0.01464(-0.02323 0.05252) | -6.23 |
| | >4 | 0.00402(0.00283 0.00521) ** | -1.71 | 0.03396(0.00069 0.06723) * | -14.46 |
| Had ANC | <4 | | | | |
| | > = 4 | -0.00283 (-0.00383–0.00183) ** | 1.21 | -0.02980(-0.06772 0.00813) | 12.68 |
| Distance to health facility | big problem | | | | |
| | not a big problem | -0.00416 (-0.00554–0.00277) ** | 1.77 | -0.02124(-0.04768 0.00520) | 9.04 |
| Total | | | 8.74 | | 91.26 |
| Constant | | | | -0.09706(-0.22455–0.03043) ** | 41.32 |

*Significance at P-valve < 0.05,

**significance at P-valve < 0.01.

ANC, antenatal care; Pct, percentage contribution

## Discussion

Gambia has established a national health policy that served as a road map from 2012–2020, with the goal of "Provision of quality and affordable Health Services for All By 2020". One of the policy's objectives was to reduce maternal mortality. There has been a lot of effort put in to achieve the targets, but maternal mortality has not decreased significantly. [25].

A characteristic effect change in women's secondary education has a significant contribution to the decline of maternal home delivery. This evidence is supported by other study findings conducted in different places. Women who complete secondary school are more likely to give birth in a health facility than women who cannot read or write [26–29]. The best reason for this is that an educated woman is less likely to be influenced by her family. Her autonomy may be great, so she may choose a health facility delivery for the benefit of her health and the new baby [30].

In Gambia, husbands' educational attainment has improved in recent years, particularly in secondary and higher education. This change in husband education played a substantial effect in the lowering of home births, and this finding is in line with others' findings [20, 31]. Male domination in Sub-Saharan Africa is enormous [32]. As a result, if the spouse is well-educated, he will make an informed choice about the delivery place. In deciding where to be delivered his baby, he may take into account the mother's and the baby's safety [33].

The number of rural residents has decreased from 52 percent in 2013 to 36.3 percent in 2019–2020, which has a big impact on the home delivery rate decrement. We can readily understand the maternal home delivery rate drops among rural residents not only due to characteristics effect change, but also change in coefficient effect (Table 4). We can explain this in another way increasing urbanization had a positive effect on health facility delivery and this result is consistent with others' studies [29, 34–36]. The possible reason will be accessibility issues, financial constraints, and poor infrastructure in rural areas may be barriers for women to giving birth in a health facility.

An increase in birth order categories proportion (two and above) harms the home delivery decrement over time and similar findings were reported [24, 34]. The rationale could be that recurrent deliveries give confidence for women that the current delivery will also end up safely at home. Another reason could be the service she obtained from her previous health facility visit did not satisfy the woman or they had a terrible experience.

There was a small increase in the number of women who had received ANC follow-up four or more times (Table 1), and this increment had a considerable impact on the lowering of maternal home births in Gambia during the two surveys time. There is well-documented evidence that women who had four or more ANC visits were more likely to give birth in a health institution than those who had fewer visits than the WHO recommendation [7, 25, 26, 29]. This could be stated that besides receiving essential services like screening for disease, identifying complications due to pregnancy, and provision of supplementary nutrition, pregnant women also received consultation regarding birth preparedness, a consequence of prolonged labor, and the advantage of giving birth in a health institute.

There has been a small change recorded in the characteristic effect of women who had no problem with access to health facilities during the survey time. This small change has a significant effect on the decrement in home delivery. The above finding is in line with the study conducted in Sub-Saharan African countries [37]. The main reason for this could be that living near a health center reduces transportation costs, makes people more aware of the benefits of giving birth in a health facility, and leads to more ANC follow-up. The coefficient effect of women giving birth in facilities varies depending on whether they had a job or not. Women who have a job are less likely to give birth at home than women who do not have a job. This

could be explained by the fact that working women are more aware of their rights and are more capable of making health decisions on their own. Furthermore, they may cover any cost relating to the delivery services, so they will not be hesitant to visit a medical institution [38].

## Conclusions

In this study, the home delivery rate had steeply decreased in Gambia during the study period of the two surveys. Just above nine-tenths decrement in home delivery rate was resulted because there was a change in the characteristics effect of the study participants between 2013 and 2019–2020 GDHS. Being urban residents and women who had occupations are variables that have a positive effect on coefficient change. The remaining decline in the home delivery rate occurred because of the difference in characteristics effect of husband education, women education, rural residents, more than three ANC follow up, and no problem reaching health facilities.

The Gambia government should take the following measures to record a further drop in home delivery, which would also provide a chance to accomplish the SDG. The first task is enhancing more citizens to attend high school and above. Secondly, narrowing the gap between rural and urban in terms of accessing health facilities, and improving the availability of infrastructure. Finally, providing health education for pregnant women through different media to have more ANC follow-ups.

## Supporting information

**S1 File.**
(DOCX)

## Acknowledgments

The author of this article would like to thank the DHS major for passing the data sets to use responsibly.

## Author Contributions

**Conceptualization:** Solomon Gedlu Nigatu.

**Data curation:** Solomon Gedlu Nigatu.

**Formal analysis:** Solomon Gedlu Nigatu.

**Funding acquisition:** Solomon Gedlu Nigatu.

**Investigation:** Solomon Gedlu Nigatu.

**Methodology:** Solomon Gedlu Nigatu.

**Project administration:** Solomon Gedlu Nigatu.

**Resources:** Solomon Gedlu Nigatu.

**Software:** Solomon Gedlu Nigatu.

**Supervision:** Solomon Gedlu Nigatu.

**Validation:** Solomon Gedlu Nigatu.

**Visualization:** Solomon Gedlu Nigatu.

**Writing – original draft:** Solomon Gedlu Nigatu.

**Writing – review & editing:** Solomon Gedlu Nigatu.

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
