## [Decision Letter · Decision Letter 0]

7 Jul 2022

PONE-D-22-03923Trend and determinants of Home Delivery in Gambia, Evidence from 2013-2020 Gambia Demographic and Health Survey: A Multivariate decomposition analysis .PLOS ONE

Dear Dr. Solomon Gedlu Nigatu,

Thank you for submitting your manuscript to PLOS ONE. After careful consideration, we feel that it has merit but does not fully meet PLOS ONE’s publication criteria as it currently stands. Therefore, we invite you to submit a revised version of the manuscript that addresses the points raised during the review process.

We look forward to receiving your revised manuscript.

Kind regards,

Wenhui Mao, PhD

Academic Editor

PLOS ONE

Journal Requirements:

3. Please include a complete copy of PLOS’ questionnaire on inclusivity in global research in your revised manuscript. Our policy for research in this area aims to improve transparency in the reporting of research performed outside of researchers’ own country or community. The policy applies to researchers who have travelled to a different country to conduct research, research with Indigenous populations or their lands, and research on cultural artefacts. The questionnaire can also be requested at the journal’s discretion for any other submissions, even if these conditions are not met.  Please find more information on the policy and a link to download a blank copy of the questionnaire here: https://journals.plos.org/plosone/s/best-practices-in-research-reporting. Please upload a completed version of your questionnaire as Supporting Information when you resubmit your manuscript.

No

No authors have competing interests

6. We note that you have indicated that data from this study are available upon request. PLOS only allows data to be available upon request if there are legal or ethical restrictions on sharing data publicly. For more information on unacceptable data access restrictions, please see http://journals.plos.org/plosone/s/data-availability#loc-unacceptable-data-access-restrictions. 

7. We note that you have stated that you will provide repository information for your data at acceptance. Should your manuscript be accepted for publication, we will hold it until you provide the relevant accession numbers or DOIs necessary to access your data. If you wish to make changes to your Data Availability statement, please describe these changes in your cover letter and we will update your Data Availability statement to reflect the information you provide.

8. Your ethics statement should only appear in the Methods section of your manuscript. If your ethics statement is written in any section besides the Methods, please delete it from any other section. 

Reviewers' comments:

Reviewer's Responses to Questions

**Comments to the Author**

1. Is the manuscript technically sound, and do the data support the conclusions?

Reviewer #1: Yes

Reviewer #2: Partly

2. Has the statistical analysis been performed appropriately and rigorously? 

Reviewer #1: Yes

Reviewer #2: No

3. Have the authors made all data underlying the findings in their manuscript fully available?

Reviewer #1: Yes

Reviewer #2: No

4. Is the manuscript presented in an intelligible fashion and written in standard English?

Reviewer #1: Yes

Reviewer #2: No

5. Review Comments to the Author

Reviewer #1: A major revision is needed on the English language and the flow information.

Discussion and conclusion sections require major revision.

The statistical methodology used in the study requires detail and clear explanation with the result/findings.

Page 2; Line 26: Gambia Demographic and Health Survey

Page 3; line 42: in this study ….

Page 6; line 120-121: dependent variable and response variable: using one of the two word is better for easily understanding and uniformity of using the term.

Page 6; line 124-125: clinics mentioned two times.

Page 6; line 124: nongovernmental hospital: is it different from private hospital. If the two are similar, cancel one of the two.

Page 7; line 127-128: move it to ethical considerations section.

Page 7; line 128-130: move it to the end of “method of data analysis” section.

Page 8; line 151: 63.7%

Page 8; line 158: study : survey

Page 12: line 247: “ ….. change in the behavior of the study participants …”. How you could proof it?

Reviewer #2: Comments to author

The subject trend and determinants of home delivery is important and likely contributor factors may differ from region to region. Thus, this might be useful study in assessing the likely predictors in the study area. However, the most important could be to assess the trend and magnitude of home delivery when home delivery was high (36.18% in 2013 GDHS) and factors associated when the home delivery rate has dropped to14.39% (in 2019-2020 GDHS). In the current form, the results do not add to the existing knowledge about the factors that are associated with home delivery. Moreover, the other issue here is a woman can give a birth at home by skilled birth attendants. So it might be very useful to assess the trend and determinants of skilled delivery instead of home delivery versus health facility delivery.

General comments: document lacking grammatical, check for sentence verb agreement, spelling and coherence. I suggest for English edition.

In the abstract and Introduction, the reason for why this study was done not convincing.

Abstract: include the odds ratios and CIs in your result section.

Line 34, what the number [34.78, 37.58] stand for? Please use statistical language to report the findings in the result section.

The introduction is not clear and difficult to follow. For the example starting from line 82, not clear what is meant “Previous research has uncovered a number of factors that contribute to the decline of home delivery” and the author mentioned some references. And starting from line 86, it stated that “Gambia has a high total fertility rate. Therefore, women gave birth to an average of 4.4 children.......” The author should rewrite the introduction so that it will be easier to follow.

Line 108, First mention the source of data. Perhaps you can change the subsection “ Study population and data collection” to Data sources and study population. First mention what is your data sources before describing the data collection produces they have used. Moreover, the dependent and independent variables should not be in the study population and data collection. They should be in a separate sub section: study variables. Line 122, the author mention that the response variable is a binary outcome, with 1 indicating that the womon gave birth at home or in another home. Do you mean at her home or another home. Please correct this. The dependent and independent variables should be in a separate subsection “study variables”. And rewrite the sentences for the dependent variable. Include the independent variables and state the variables in detail.

In the study area, it is important to describe the study area in more detail. For example the estimated populations living in rural and urban areas, and the number of health facilities in the region.

The method section is difficult to follow. The author should rewrite the method section so that it will be easiy to follow. For example method for data analysis the author should focose on the techniques that used to address the objective, instead of discussing the meaning of confidence intervals.

The result section is difficult to follow. Please consider rewording the paragraphs to make it easier to follow.

Line 151, change the value 97% to 98%.

Lines 158-160: Lack clarity, not clear what is meant by “Home delivery was 36.18 % [34.78, 37.58] in Gambia in the first study. After five years, another survey was conducted in 2019-2020...”

Grammatical errors are present sporadically across the write-up. English needs to be corrected.

If the data is reanalyzed in light of the above suggestion, the discussion need to be aligned with the revised analysis and results. The findings should be sufficiently discussed in comparison with previous research studies and the existing differences should be fully clarified. For example, what is the new finding which differ from existing literatures. Currently the discussion does not discuss in depth to add to the existing knowledge about factors that determine access to health faculties delivery. Discussion needs to be strengthened..

6. PLOS authors have the option to publish the peer review history of their article (what does this mean?). If published, this will include your full peer review and any attached files.

Reviewer #1: No

Reviewer #2: No

---

## [Author Response · Author response to Decision Letter 0]

7 Oct 2022

Date: September 17, 2022

To: “PLOS ONE Editorial Office" PONE-D-22-03923

From: "Solomon Gedlu" sol.gondar@gmail.com

Subject: Author’s response to reviewers’ comments 

Trend and determinants of Home Delivery in Gambia, Evidence from 2013-2020 Gambia Demographic and Health Survey: A Multivariate decomposition analysis.

Solomon Gedlu1 

1Department of Epidemiology and Biostatistics, Institute of Public Health, University of Gondar, Gondar, Ethiopia. 

Dear Editor,

We would like to express my appreciation for the pertinent comments and helpful suggestions from the reviewers on my manuscript. Based on the reviewers’ comments and suggestions, I further revised the manuscript and made corrections and modifications accordingly. Below are attached my point-by-point responses to the comments and suggestions. I hope that the revised manuscript address fully the reviewers’ comments. Finally, appreciate your kind suggestion and comments on my work manuscript.

Sincerely yours, 

Solomon Gedlu Nigatu, author

Point-by-point response

Response to Reviewer #1

Comment 1: A major revision is needed on the English language and the flow information.

Author response 1: Dear reviewer thank you so much for your constructive comment. As you saw and I also checked the write-up had a huge language error. I took a series of measures to correct all language related issues.

Comment 2: Discussion and conclusion sections require major revision.

Author response 2: Dear reviewer thank you so much. As per your request and I checked not only the mentioned parts but also additional parts of the manuscript that had unclarity. I hope the revised looks good to the reader.

Comment 3: The statistical methodology used in the study requires detail and clear explanation with the result/findings.

Author response 3: Dear reviewer thank you so much for your valuable comment. Yes, to make the paper more concise I missed some important information which deal about the analysis I used. On the revised manuscript I did my best to explain what does mean decomposition analysis and how the analysis was done. 

Comment 4: Page 2; Line 26: Gambia Demographic and Health Survey

Author response 4: Dear reviewer thank you so much. I corrected it. 

Comment 5: Page 3; line 42: in this study ….

Author response 5: Dear reviewer thank you so much for your suggestion. I made corrections as per the suggestion.

Comment 6: Page 6; line 120-121: dependent variable and response variable: using one of the two word is better for easily understanding and uniformity of using the term.

Author response 6: Dear reviewer thank you so much. I used these two words interchangeably but to avoid confusion for the readers I used the dependent variable.

Comment 7: Page 6; line 124-125: clinics mentioned two times.

Author response 7: Dear reviewer thank you so much. I deleted it. 

Comment 8: Page 6; line 124: nongovernmental hospital: is it different from private hospital. If the two are similar, cancel one of the two.

Author response 8: Dear reviewer thank you so much for your curiosity question. There is no difference between the hospitals regarding the delivery service. So, I mentioned only private hospitals.

Comment 9: Page 7; line 127-128: move it to ethical considerations section.

Author response 9: Dear reviewer thank you so much. I did as per your recommendation.

Comment 10: Page 7; line 128-130: move it to the end of “method of data analysis” section.

Author response 10: Dear reviewer thank you so much. Your recommendation sounds, so I did it.

Comment 11: Page 8; line 151: 63.7%

Author response 11: Dear reviewer thank you so much. I missed the percentage. Now it is corrected.

Comment 12: Page 8; line 158: study: survey

Author response 12: Dear reviewer thank you for understanding me. It was wrong to say study. On the revised it is correct.

Comment 13: Page 12: line 247: “ ….. change in the behavior of the study participants …”. How you could proof it?

Author response 13: Dear reviewer thank you so much for your curiosity question. In decomposition analysis, if the trend is significant, the contributor variables for the change are either characteristic effect or coefficient effect. Sometimes both will contribute to the change. Keep in mind this, the coefficient effect can be mentioned using different terms like effect, coefficients, and behavioral. In the revised manuscript I used the term coefficient effect consistently.

Response to Reviewer #2

Comment 1: The subject trend and determinants of home delivery is important and likely contributor factors may differ from region to region. Thus, this might be useful study in assessing the likely predictors in the study area. However, the most important could be to assess the trend and magnitude of home delivery when home delivery was high (36.18% in 2013 GDHS) and factors associated when the home delivery rate has dropped to14.39% (in 2019-2020 GDHS). In the current form, the results do not add to the existing knowledge about the factors that are associated with home delivery. Moreover, the other issue here is a woman can give a birth at home by skilled birth attendants. So it might be very useful to assess the trend and determinants of skilled delivery instead of home delivery versus health facility delivery.

General comments: document lacking grammatical, check for sentence verb agreement, spelling and coherence. I suggest for English edition.

Author response1: Dear reviewer thank you so much for your constructive comment. In Gambia maternal mortality ratio is still high. Even though the data set does not clearly show how that specific home birth assisted, birth takes place at home will not be supported by a skilled professional. I believe this manuscript will contribute to the knowledge as well as indicate possible solutions for the problem. For sure there were tremendous grammatical, spelling, and coherence problems. The author did his all capacity for English edition. I hope the revised edition meet he journal requirement for language clarity.

Comment 2: In the abstract and Introduction, the reason for why this study was done not convincing.

Author response 2: Dear reviewer thank you so much for your valuable comment. In the revised version more detail regarding the rationale of the study is stated. Please have a look.

Comment 3: Abstract: include the odds ratios and CIs in your result section.

Author response3: Dear reviewer thank you so much for your suggestion. In a decomposition analysis odds ratio cannot do anything rather the confidence interval with 95% is the most important. 

Comment 4: Line 34, what the number [34.78, 37.58] stand for? Please use statistical language to report the findings in the result section. 

Author response 4: Dear reviewer thank you so much for your comment. Sorry for the unclarity happened. It is a confidence interval but it missed CI and 95 %. It is corrected accordingly.

Comment 5: The introduction is not clear and difficult to follow. For the example starting from line 82, not clear what is meant “Previous research has uncovered a number of factors that contribute to the decline of home delivery” and the author mentioned some references. And starting from line 86, it stated that “Gambia has a high total fertility rate. Therefore, women gave birth to an average of 4.4 children.......” The author should rewrite the introduction so that it will be easier to follow. 

Author response 5: Dear reviewer thank you so much for your comment. The paragraph has a grammatical error. So, it was rewritten to give a clear message.

Comment 6: Line 108, First mention the source of data. Perhaps you can change the subsection “ Study population and data collection” to Data sources and study population. First mention what is your data sources before describing the data collection produces they have used. Moreover, the dependent and independent variables should not be in the study population and data collection. They should be in a separate sub section: study variables. Line 122, the author mention that the response variable is a binary outcome, with 1 indicating that the womon gave birth at home or in another home. Do you mean at her home or another home. Please correct this. The dependent and independent variables should be in a separate subsection “study variables”. And rewrite the sentences for the dependent variable. Include the independent variables and state the variables in detail.

Author response 6: Dear reviewer thank you so much for your constructive comments. It is stated as "Data source, Study population, and sampling technique” in the revised manuscript. I also put the study variable in a sub section as dependent and independent variables and encompassed all the details. The phrase “woman gave birth at home or in another home” I wanted to say in her home and correction is done on it.

Comment 7: In the study area, it is important to describe the study area in more detail. For example the estimated populations living in rural and urban areas, and the number of health facilities in the region.

Author response 7: Dear reviewer thank you so much for your constructive comments. Yes, the most important information was missed. In the revised manuscript all the requested information is included. 

Comment 8: The method section is difficult to follow. The author should rewrite the method section so that it will be easiy to follow. For example method for data analysis the author should focose on the techniques that used to address the objective, instead of discussing the meaning of confidence intervals.

Author response 8: Dear reviewer thank you so much for your constructive comments. I also checked it and it needs much improvement. So, I did a lot of effort to make it easily understandable to the readers. 

Comment 9:The result section is difficult to follow. Please consider rewording the paragraphs to make it easier to follow.

Author response 9: Dear reviewer thank you so much for your constructive comments. The same is true for this section like the method part. I hope now it sounds.

Comment 10: Line 151, change the value 97% to 98%.

Author response 10: Dear reviewer thank you so much. It should be 97.9%. Now the error is fixed. 

Comment 11: Lines 158-160: Lack clarity, not clear what is meant by “Home delivery was 36.18 % [34.78, 37.58] in Gambia in the first study. After five years, another survey was conducted in 2019-2020...”

Author response 11: Dear reviewer thank you so much for your constructive comments. The author wanted to mention the point estimation of home delivery with its respective confidence interval. 

Comment 12: Grammatical errors are present sporadically across the write-up. English needs to be corrected.

Author response 12: Dear reviewer thank you so much for your constructive comment. Yes, indeed there were several grammatic errors as explained previously, I did all effort to overcome such a problem.

Comment 13: If the data is reanalyzed in light of the above suggestion, the discussion need to be aligned with the revised analysis and results. The findings should be sufficiently discussed in comparison with previous research studies and the existing differences should be fully clarified. For example, what is the new finding which differ from existing literatures. Currently the discussion does not discuss in depth to add to the existing knowledge about factors that determine access to health faculties delivery. Discussion needs to be strengthened..

Author response 13: Dear reviewer thank you so much for your constructive comment. I do not think there is a need to have a reanalysis. I strongly believed the analysis was well done but the way I write was not good enough. So, the revised manuscript considered all the comments including language. If there is a need to improve, I am open to doing so.

---

## [Decision Letter · Decision Letter 1]

26 Jul 2023

PONE-D-22-03923R1Trend and determinants of Home Delivery in Gambia, Evidence from 2013-2020 Gambia Demographic and Health Survey: A Multivariate decomposition analysis .PLOS ONE

Dear Dr. Nigatu,

Thank you for submitting your manuscript to PLOS ONE. After careful consideration, we feel that it has merit but does not fully meet PLOS ONE’s publication criteria as it currently stands. Therefore, we invite you to submit a revised version of the manuscript that addresses the points raised during the review process.

Dear Dr. Gedlu Solomon,

Thank you for submitting your manuscript to PLOS ONE. After careful consideration, we feel that it has merit but does not fully meet PLOS ONE’s publication criteria as it currently stands. Therefore, we invite you to submit a revised version of the manuscript that addresses the points raised during the review process. Please submit your revised manuscript by Sep 09 2023 11:59PM. If you will need more time than this to complete your revisions, please reply to this message or contact the journal office at plosone@plos.org. **Comments**

Generally the manuscript is good. But, it needs an edition by English language professionals. Besides, you should answer for all the questions raised by reviewers.

We look forward to receiving your revised manuscript.

Kind regards.

Hassen Mosa, MSc

Academic Editor

PLOS ONE

We look forward to receiving your revised manuscript.

Kind regards,

Hassen Mosa, Msc

Academic Editor

PLOS ONE

Journal Requirements:

Reviewers' comments:

Reviewer's Responses to Questions

**Comments to the Author**

1. If the authors have adequately addressed your comments raised in a previous round of review and you feel that this manuscript is now acceptable for publication, you may indicate that here to bypass the “Comments to the Author” section, enter your conflict of interest statement in the “Confidential to Editor” section, and submit your "Accept" recommendation.

Reviewer #1: All comments have been addressed

Reviewer #3: (No Response)

2. Is the manuscript technically sound, and do the data support the conclusions?

Reviewer #1: Yes

Reviewer #3: No

3. Has the statistical analysis been performed appropriately and rigorously? 

Reviewer #1: Yes

Reviewer #3: Yes

4. Have the authors made all data underlying the findings in their manuscript fully available?

Reviewer #1: Yes

Reviewer #3: No

5. Is the manuscript presented in an intelligible fashion and written in standard English?

Reviewer #1: Yes

Reviewer #3: No

6. Review Comments to the Author

Reviewer #1: All review comments and suggestions have been addressed by the author.

The subject trend and determinants of home delivery is important, and likely contributor factors may differ from region to region. Thus, this might be a useful study in assessing the likely predictors in the study area.

Reviewer #3: Trend and determinants of Home Delivery in Gambia, Evidence from 2013-2020

Gambia Demographic and Health Survey: A Multivariate decomposition analysis

The author did a good job in this document but the document is far from acceptable academic writing.

Abstract

- At the onset, home delivery should be clearly and explicitly defined, which is not the case. I suggest something like defining the main variable, describing its severity, making a connection with Gambia, and the gap, and showing how your objective will solve the gap.

- The current introduction does not make any sense in an academic writing context and should be rewritten.

- Look at the first two sentences, they are just connected and are not clear

- Overall, the abstract is good but confusing and does not leave any impressive impression. I suggest redrafting it according to the following guidelines. The first two lines must encompass the context of the study and the research problem, further two lines must be covered the objective of the papers with unfolding the description of the title. In the next 2 to 4 lines the methodology will be covered. Afterward, the next two lines are for result and performance. In these lines, the author must define how the results and performance are being achieved, for instance, by conducting either simulation or physical implementation. Please mention the name of the simulation or the physical method. The result statistics must be mentioned in the last two lines and either in percentage or with real-time values.

Introduction

- The authors must give a clear procedure for the proposed solution with an algorithm (via flowcharts or pseudo codes, i.e. the flowcharts of a proposal work must be drawn), and must be supported with a figure either a block diagram of the proposed methodology or the topology in a formal style.

- In the Literature Review section, the authors must cover the shortcomings because shortcoming or challenges in previous work is not mentioned. If the previous work is free of challenges and there is no issue then what motivated the authors to propose this study? It inculcates that authors should revise the literature review and critically highlight the problems in the previous study and compare the proposed solution and tells how the proposed solution is best fitted.

- And please, give special attention to writing English. There are many switching tenses, punctuation errors, poor word choices, and unclear and complicated wordy sentences throughout the document.

- You may add a table and present the summary of related work, the shortcomings, and the proposed solution accordingly.

Methods

- In the Research Methodology section, what is the difference between your proposed method and other based techniques? (II) Please compare your proposed method with other based methods in terms of time complexity.

Results

- Critical analysis of the finding, which is the most important part, is missing. This would help the readers to further improve the study.

- Again, go for a thorough proofread of the paper to rectify several existing typos and grammatical mistakes to improve the written quality of the paper. If necessary, take the help of a native English speaker to improve the language of the paper.

-

Conclusion

- Regarding the conclusion paragraph, Please precisely describe the outcome of the study and justify the statements that are mentioned in the abstract. Further, it must contain additional points and must give a clear and more discussion about the experimental results. The main novelty and contribution of needs must be summarized and highlight the recommendations based on obtained results. These results are the hallmark for future extension therefore, please spend some more time writing the conclusion and based on the results suggest new directions.

References

While reviewing the references, I observed that cited references are outdated whereas more work already has been done in the proposed study. The cited references are neither sufficient nor suitable and therefore must extend the list and focus only on the papers from the recent 3 years. The following references may be added to supersede the outdated ones for the authors' convenience.

7. PLOS authors have the option to publish the peer review history of their article (what does this mean?). If published, this will include your full peer review and any attached files.

Reviewer #1: No

Reviewer #3: No

---

## [Author Response · Author response to Decision Letter 1]

9 Sep 2023

Date: September 9, 2023

To: PLOS ONE Editorial Office. manuscript ID: PONE-D-22-03923

From: "Solomon Gedlu" sol.gondar@gmail.com

Subject: Author’s response to reviewers’ comments 

Trend and determinants of Home Delivery in Gambia, Evidence from 2013 and 2020 Gambia Demographic and Health Survey: A Multivariate Decomposition Analysis.

Solomon Gedlu1 

1Department of Epidemiology and Biostatistics, Institute of Public Health, University of Gondar, Gondar, Ethiopia. 

Dear Editor,

I would like to express my appreciation for all who participant in the review process of my manuscript. Based on the reviewers’ comments and suggestions, I revised it again and made corrections accordingly. I hope that the revised manuscript address fully the reviewers’ comments. I am always willing to improve the manuscript, so please do not hesitate to forward your feedback if there is anything I missed.

Best regards, 

Solomon Gedlu Nigatu, author

Point-by-point response

Response to Reviewer #3

 Comment: At the onset, home delivery should be clearly and explicitly defined, which is not the case. I suggest something like defining the main variable, describing its severity, making a connection with Gambia, and the gap, and showing how your objective will solve the gap.

Author response: Dear reviewer thank you for your valuable comment. As per the comment and suggestion I incorporate the definition of what does mean home delivery and its magnitude on different scale. All the change is available on revised manuscript. 

Comment 1: -The current introduction does not make any sense in an academic writing context and should be rewritten. Look at the first two sentences, they are just connected and are not clear. Overall, the abstract is good but confusing and does not leave any impressive impression. I suggest redrafting it according to the following guidelines. The first two lines must encompass the context of the study and the research problem, further two lines must be covered the objective of the papers with unfolding the description of the title. In the next 2 to 4 lines the methodology will be covered. Afterward, the next two lines are for result and performance. In these lines, the author must define how the results and performance are being achieved, for instance, by conducting either simulation or physical implementation. Please mention the name of the simulation or the physical method. The result statistics must be mentioned in the last two lines and either in percentage or with real-time values.

Author response: Thank you dear reviewer for your constructive comment. As per the recommendation I made a change on the introduction part of the manuscript. However, Point raised to include simulation or physical implementation is not possible for such kind study. As to my best this is a epidemiological cross sectional study rather than implantation study. 

Comment 1: -The authors must give a clear procedure for the proposed solution with an algorithm (via flowcharts or pseudo codes, i.e. the flowcharts of a proposal work must be drawn), and must be supported with a figure either a block diagram of the proposed methodology or the topology in a formal style.

Author response: Dear reviewer thank you for your valuable comment. This is a health science study to figure what the extent of the public health problem and to know the contributor factors. What dear reviewer recommend like flowcharts or pseudo code may suitable for computer science studies.

Comment 1: - In the Literature Review section, the authors must cover the shortcomings because shortcoming or challenges in previous work is not mentioned. If the previous work is free of challenges and there is no issue then what motivated the authors to propose this study? It inculcates that authors should revise the literature review and critically highlight the problems in the previous study and compare the proposed solution and tells how the proposed solution is best fitted.

Author response: Thank you dear reviewer for your constructive comment. As far as I know there is no literature review section, so its difficult for me to address what you mention.

Comment 1: And please, give special attention to writing English. There are many switching tenses, punctuation errors, poor word choices, and unclear and complicated wordy sentences throughout the document.

Author response: Dear reviewer thank you for your valuable comment. You are right there were a number of grammar and punctuation errors. My colleagues and I have strived to avoid the mistakes. 

Comment 1: You may add a table and present the summary of related work, the shortcomings, and the proposed solution accordingly.

Author response: Dear reviewer thank you for your valuable comment. This is basic research for the seek of advancing knowledge on human health issue. For applied (operation) research there a solution and you test the effectives of the proposed solution. For my case it does not work.

Comment 1: -In the Research Methodology section, what is the difference between your proposed method and other based techniques? (II) Please compare your proposed method with other based methods in terms of time complexity.

Author response: Dear reviewer thank you for your valuable comment. Please view my previous response.

Comment 1: Critical analysis of the finding, which is the most important part, is missing. This would help the readers to further improve the study.

Author response: Dear reviewer thank you for your valuable comment. For this study I used a decomposition analysis and it presented what does it mean, the formula and how it will interpret in the method part. If there is any thing miss, I am willing to address it again. 

Comment 1: Again, go for a thorough proofread of the paper to rectify several existing typos and grammatical mistakes to improve the written quality of the paper. If necessary, take the help of a native English speaker to improve the language of the paper.

Author response: Thank you dear reviewer for your constructive comment. It addressed well with the help of my colleagues

Comment 1: Regarding the conclusion paragraph, please precisely describe the outcome of the study and justify the statements that are mentioned in the abstract. Further, it must contain additional points and must give a clear and more discussion about the experimental results. The main novelty and contribution of needs must be summarized and highlight the recommendations based on obtained results. These results are the hallmark for future extension therefore, please spend some more time writing the conclusion and based on the results suggest new directions.

Author response: Dear reviewer thank you for your valuable comment. The general objective of the study was to assess the trend of home delivery and identify factors. As far as I concern the outcome of the study will be what is the trend of home delivery “steeply declined” and I mentioned factors contributed for such decline.

Comment 1: While reviewing the references, I observed that cited references are outdated whereas more work already has been done in the proposed study. The cited references are neither sufficient nor suitable and therefore must extend the list and focus only on the papers from the recent 3 years. The following references may be added to supersede the outdated ones for the authors' convenience.

Author response. Dear reviewer thanks you for your valuable comment. I tried to incorporate those articles published in last five years. It is hard to get articles published within three years.

---

## [Decision Letter · Decision Letter 2]

20 Nov 2023

Trend and determinants of Home Delivery in Gambia, Evidence from 2013 and 2020 Gambia Demographic and Health Survey: A Multivariate Decomposition Analysis .

PONE-D-22-03923R2

Dear Dr. Gedlu S,We’re pleased to inform you that your manuscript has been judged scientifically suitable for publication and will be formally accepted for publication once it meets all outstanding technical requirements.

Kind regards,

Hassen Mosa, Msc

Academic Editor

PLOS ONE

Additional Editor Comments (optional):

Reviewers' comments:

Reviewer's Responses to Questions

**Comments to the Author**

1. If the authors have adequately addressed your comments raised in a previous round of review and you feel that this manuscript is now acceptable for publication, you may indicate that here to bypass the “Comments to the Author” section, enter your conflict of interest statement in the “Confidential to Editor” section, and submit your "Accept" recommendation.

Reviewer #3: All comments have been addressed

Reviewer #4: All comments have been addressed

2. Is the manuscript technically sound, and do the data support the conclusions?

Reviewer #3: Yes

Reviewer #4: Yes

3. Has the statistical analysis been performed appropriately and rigorously? 

Reviewer #3: Yes

Reviewer #4: Yes

4. Have the authors made all data underlying the findings in their manuscript fully available?

Reviewer #3: No

Reviewer #4: Yes

5. Is the manuscript presented in an intelligible fashion and written in standard English?

Reviewer #3: Yes

Reviewer #4: Yes

6. Review Comments to the Author

Reviewer #3: please read over again improve few grammars error that still exist. Add one sentence in abstract introduction before purpose statement (Therefore...) that state a possible solution for the stated problem.

Reviewer #4: This manuscript rises timely health agenda and it has written best way. it has no english grammar error and other problem.

7. PLOS authors have the option to publish the peer review history of their article (what does this mean?). If published, this will include your full peer review and any attached files.

Reviewer #3: **Yes: **Girma Gilano

Reviewer #4: **Yes: **Ayele Mamo Abebe

---

## [Editor Report · Acceptance letter]

24 Nov 2023

PONE-D-22-03923R2 

Trend and determinants of Home Delivery in Gambia, Evidence from 2013 and 2020 Gambia Demographic and Health Survey: A Multivariate Decomposition Analysis. 

Dear Dr. Nigatu:

I'm pleased to inform you that your manuscript has been deemed suitable for publication in PLOS ONE. Congratulations! Your manuscript is now with our production department. 

Kind regards, 

on behalf of

Mr Hassen Mosa 

Academic Editor

PLOS ONE